# GRI: Graph-based Relative Isomorphism of Word Embedding Spaces

**Muhammad Asif Ali,**[1] **Yan Hu,**[1] **Jianbin Qin,**[2] **Di Wang**[1]

[1] King Abdullah University of Science and Technology, KSA

[2] Shenzhen University, China

{muhammadasif.ali, yan.hu, di.wang}@kaust.edu.sa, qinjianbin@szu.edu.cn

## Abstract

Automated construction of bilingual dictionaries using monolingual embedding spaces is a core challenge in machine translation. The end performance of these dictionaries relies upon the geometric similarity of individual spaces, i.e., their degree of isomorphism. Existing attempts aimed at controlling the relative isomorphism of different spaces fail to incorporate the impact of semantically related words in the training objective. To address this, we propose GRI that combines the distributional training objectives with attentive graph convolutions to unanimously consider the impact of semantically similar words required to define/compute the relative isomorphism of multiple spaces. Experimental evaluation shows that GRI outperforms the existing research by improving the average P@1 by a relative score of up to 63.6%. We release the codes for GRI at https://github.com/asif6827/GRI.

## 1 Introduction

Bilingual Lexical Induction (BLI) aims at the construction of lexical dictionaries using different mono-lingual word embeddings. Automated construction of bilingual dictionaries plays a significant role, especially for resource-constrained languages where hand-crafted dictionaries are almost non-existent. It is also a key tool to bootstrap the performance of many down-streaming applications, e.g., cross-lingual information retrieval (Artetxe et al., 2018a), neural machine translation (Lample et al., 2018).

The most prevalent way for the construction of cross-lingual embeddings is to map the monolingual embeddings in a shared space using linear and/or non-linear transformations, also known as mapping-based methods (Conneau et al., 2017; Joulin et al., 2018; Patra et al., 2019). A core limitation of the mapping-based methods is their reliance on the approximate isomorphism assumption, i.e., the underlying monolingual embedding spaces are

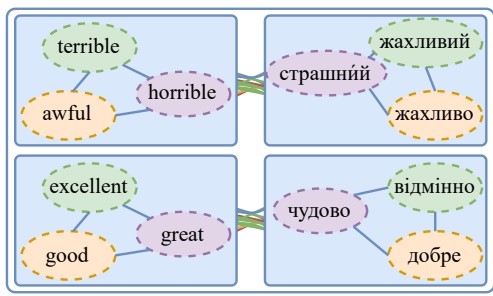

Figure 1: Semantically related tokens for English and Ukrainian languages. These words though lexically varying carry the same semantics and their impact should be unanimously considered.

geometrically similar. This severely limits the applicability of the mapping-based methods to closely related languages and similar data domains. This isomorphism assumption does not hold, especially in case of domain-mismatch and for languages exhibiting different characteristics (Conneau et al., 2017; Søgaard et al., 2018; Glavas et al., 2019; Patra et al., 2019). Other dominant factors identified in the literature that limit the end performance of BLI systems include: (i) linguistic differences (ii) algorithmic mismatch, (iii) variation in data size, (iv) parameterization etc. Similar to the supervised models, the unsupervised variants of BLI are also unable to cater to the above-mentioned challenges (Kim et al., 2020; Marie and Fujita, 2020). Instead of relying on embedding spaces trained completely independent of each other, in the recent past there have been a shift in explicitly using the isomorphism measures alongside distributional training objective (Marchisio et al., 2022). In order to control the relative isomorphism of monolingual embedding spaces, these models use existing bilingual dictionaries as training seeds. However, one core limitation of these models is their inability to incorporate the impact of semantically relevant tokens into the training objective. This severely deteriorates the relative isomorphism of the resultant cross-lingual embedding space.

This phenomenon is illustrated in Figure 1 for English and Ukrainian languages. For example, for the English language, we often use terms {*"terrible", "horrible"*} within the same context without a significant change in the meaning of the sentence. For these terms, corresponding terms in the Ukrainian language {"страшний", "жахливо"} may also be used interchangeably without a significant change in the context. Likewise, for the bottom row in Figure 1, the words {*"good", "great", "excellent"*} are semantically related words in the English language, with {"відмінно", "чудово", "добре"} as corresponding semantically related words in the Ukrainian language.

To address these challenges, in this paper we propose a novel framework named: Graph-based Relative Isomorphism (GRI). GRI uses attentive graph convolutions to pay attention to semantically related tokens, followed by using isomorphism metrics to inject this information into the model training. Later, it combines the isomorphism loss with the distributional training objective to train the complete model.

We argue GRI offers a better alternative for BLI, as it allows injecting information about the semantic variations of tokens in the training objective, which is a more natural setting in order to control the relative isomorphism of linguistic data. An immediate benefit of the proposed model is obvious in the domain-mismatch settings, where attentive graph convolutions mechanism by GRI offer the ability to unanimously analyze and/or model similar concepts represented by lexically varying terms across different corpora. This is also evident by a relatively stable performance of GRI for both domain-sharing and domain-mismatch settings (Section 6.1). We summarize the core contributions of this paper as follows:

1. We propose GRI that combines isomorphism loss functions (guided by graph convolutions) along with the distributional training objective for BLI.

2. We propose attentive graph convolutions for GRI in order to control the relative isomorphism by sharing information across semantically related tokens.

3. We illustrate the effectiveness of GRI via comprehensive experimentation. For benchmark data sets, GRI outperforms the existing state of the art by approximately 63.6% for average P@1.

## 2 Related Work

Due to limited space, we primarily categorize the related work on relative isomorphism of cross-lingual embeddings into: (i) mapping to shared space, and (ii) joint training.

**Mapping to shared space.** These models aim to find a linear and/or non-linear transformation for pre-trained word embeddings in order to map them to a shared space. These models rely on the assumption that the embedding models share similar structure across different languages (Mikolov et al., 2013a), which allows them to independently train embeddings for different languages and learn mapping functions to align them in a shared space. Supervised variants in this regard use existing bilingual resources, such as parallel dictionaries (Xing et al., 2015; Joulin et al., 2018; Jawanpuria et al., 2019). The unsupervised variants use distributional matching (Zhang et al., 2017; Conneau et al., 2017; Artetxe et al., 2018b; Zhou et al., 2019). These models have also been applied to the contextualized embeddings (Aldarmaki and Diab, 2019; Schuster et al., 2019).

**Joint Training** These models put additional constraints on model learning, i.e., a hard or soft cross-lingual constraints in addition to the monolingual training objectives. Similar to the mapping-based models, early works in this domain include the supervised variants relying on bilingual dictionaries (Ammar et al., 2016; Luong et al., 2015; Gouws et al., 2015). Recently, the unsupervised approaches have gained attention because of their ease of implementation. For instance, Lample et al. (2018) analyzed the performance for concatenated monolingual corpora with shared vocabulary without any additional cross-lingual resources. Results show that this setting outperforms many carefully crafted alignment based strategies for unsupervised machine translation. Other unsupervised approaches with good results on benchmark data sets include zero-shot cross-lingual transfer by Artetxe and Schwenk (2019) and cross-lingual pre-training by Lample and Conneau (2019). Marchisio et al. (2022) proposed IsoVec that introduces multiple different losses along with the skip-gram loss function to control the relative isomorphism of monolingual spaces.

A major limitation of these methods is their inability to incorporate the lexico-semantic variations of word pairs across different languages in the model training, which severely limits the end performance of these models.

## 3 Background

In this section, we discuss the notation and mathematical background of the tools and techniques used in this paper.

### 3.1 Notation

Throughout this paper, we use $\mathbf{U} \in \mathbf{R}^{p \times d}$ and $\mathbf{V} \in \mathbf{R}^{q \times d}$ to represent the embeddings of the source and target languages. We assume the availability of seeds pairs for both source and target languages, represented by: $\{(u_0, v_0), (u_1, v_1), ...(u_s, v_s)\}$.

### 3.2 VecMap toolkit

For mapping across different embedding spaces, we use vecmap toolkit[1]. We follow Zhang et al. (2019) to pre-process the embeddings, i.e., the embeddings are unit-normed, mean-centered and unit-normed again. For bilingual induction, we follow the steps outlined by (Artetxe et al., 2018a), i.e., whitening each space, and solving Procrustes. This is followed by re-weighting, de-whitening, and mapping of translation pairs via nearest-neighbor retrieval. For details, refer to the original work by Artetxe et al. (2018a).

## 4 Proposed Approach

### 4.1 Problem Definition

In this paper, we address a core challenge in controlling the relative isomorphism for cross-lingual data sets, i.e., incorporate the impact of semantically coherent words for BLI.

### 4.2 Overview

We propose Graph-based Relative Isomorphism GRI, shown in Figure 2, that aims to learn distributional information in the source embedding space $\mathbf{U}$, in such a way: (i) $\mathbf{U}$ is geometrically similar to the target embedding space $\mathbf{V}$ to the best possible extent, (ii) $\mathbf{U}$ captures information about the semantically related terms in $\mathbf{V}$. In order to capture the distributional information GRI uses skip-gram with negative sampling. In order to control the geometry and isomorphism of embedding space

[1] https://github.com/artetxem/vecmap

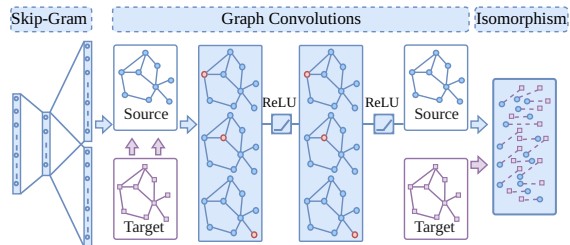

Figure 2: Proposed framework for Graph-based Relative Isomorphism(GRI). It combines attentive graph convolutions with the skip-gram to control the relative isomorphism for source $\mathbf{U}$ and target $\mathbf{V}$ embeddings.

$\mathbf{U}$ relative to space $\mathbf{V}$, GRI uses attentive graph convolutions. Finally, it uses multiple different isomorphism metrics along with the skip-gram loss function for model training.

We claim the proposed model provides the provision to perform BLI in a performance-enhanced fashion by using attentive graph convolutions for effective propagation of semantic relatedness of tokens across different languages.

### 4.3 GRI

In order to learn the distributional embeddings for the source language that are geometrically similar to the target embeddings GRI incorporates attentive graph convolutions along with the distributional training objective. GRI relies on the assumption that each language possesses multiple variations of semantically similar tokens that may be used interchangeably. And, in order to effectively model the relative isomorphism for the multi-lingual data sets this phenomenon needs to be captured explicitly.

The proposed model (GRI) is based on the assumption that sharing information among semantically related words is inevitable in order to control the relative isomorphism of the embedding spaces. From the linguistic perspective, there are an arbitrary number of words semantically related to a given word, which makes graphs a natural choice to unanimously consider the impact of these words in the end model. We explain the individual components of GRI in the following sub-sections:

#### 4.3.1 Distributional Representation

In order to learn the distributional representations GRI uses skip-gram with negative sampling. For training the skip-gram model, we follow the same settings as outlined by Mikolov et al. (2013b), i.e., embed a word close to its contextual neighbors and far from a set of words randomly chosen from the vocabulary. The formulation of the skip-gram loss function is illustrated in Equation 1.

$$\mathcal{L}_{SG} = \log \sigma(u'_{c_O}{}^{\mathsf{T}} u_{c_I}) +$$
$$\sum_i^k \mathbf{E}_{c_i \sim P_n(c)} \big[ \log \sigma(-u'_{c_i}{}^{\mathsf{T}} u_{c_I}) \big] \tag{1}$$

Here $u_{c_O}$ and $u_{c_I}$ correspond to the output and input vector representation of the word $c$. $u'_{c_i}$ the embedding vectors for noise terms. $P_n(c)$ corresponds to the noise distribution, $k$ is the number of noisy samples. We use $k = 10$ in our case.

### 4.3.2 Capturing Semantics

In order to control the relative isomorphism across the source and target embeddings GRI uses attentive graph convolutions under transductive settings in order to share information among semantically related words. The graph construction is summarized in Algorithm 1, and explained as follows:

**Graph Construction.** Inputs for the graph construction include: (i) the supervision seed pairs for the target language, (ii) existing pre-trained word2vec embeddings[2] (Mikolov et al., 2013b). The graph construction process proceeds as follows:

Firstly, we organize the target words into all possible pairs, i.e., combinations of two words at a time. For each word pair, we compute a score (cosine similarity) of their corresponding embedding vectors. The word pairs with scores higher than a threshold ($thr$) are stored as the probable semantically related terms (Pairs$_{prob}$), illustrated in lines (2-6). We observed that using a significantly higher value for $thr$ is beneficial, because: (i) it helps in capturing the most confident word pairs thus overcoming noise, and (ii) only a few word pairs end up in Pairs$_{prob}$ which makes it computationally efficient. Finally, for all word pairs in Pairs$_{prob}$, we formulate edges to construct the graph G. For each word pair, we use the cosine score of corresponding embedding vectors as the attention weight.

**Attentive Graph Convolutions.** Depending upon the value of $thr$, graph G surrounds each word by a set of highly confident semantically related words (including their lexical variations). The degree of similarity is controlled by the cosine similarity of embedding vectors. Later, for each word, we aggregate the information contained in the neighbors to come up with a new representation

[2] https://code.google.com/archive/p/word2vec/, trained using Google-News Corpus of 100 billion words.

---

**Algorithm 1** Graph Construction
**Input:** Embedding (EMB);
$\quad\quad$ $D_{tar} = \text{Target}(D_{tr+dev+tst})$
**Output:** Graph: G
1: Pairs$_{prob} \leftarrow \mathbf{0}$; G $\leftarrow \emptyset$
2: **for** $(w_1, w_2) \leftarrow \text{Pairs}(D_{tar})$ **do**
3: $\quad\quad$ $y^* = \text{score}_{\text{EMB}}(w_1, w_2)$
4: $\quad\quad$ **if** $y^* \geq thr$ **then**
5: $\quad\quad\quad\quad$ Pairs$_{prob} \leftarrow$ Pairs$_{prob} \cup (w_1, w_2)$
6: $\quad\quad$ **end if**
7: **end for**
8: **for** $pair \in$ Pairs$_{prob}$ **do**
9: $\quad\quad$ G $\leftarrow$ G $\cup \{edge(pair)\}$
10: **end for**
11: **return** G

---

of the word that accommodates information from semantically related neighbors.

Note, in our setting, unlike the existing work by Kipf and Welling (2016), we propose attentive graph convolutions with pair-wise distributional similarity scores as the hard attention weights. The attention weights are not updated during the model training. Specifically, we use the following layer-wise propagation mechanism:

$$L^{(i+1)} = \rho(\tilde{\Gamma} L^{(i)} W_i) \tag{2}$$

where $\tilde{\Gamma} = \bar{D}^{-1/2}(\Gamma + I)\bar{D}^{-1/2}$ is the normalized symmetric matrix, $\bar{D}$ is the degree matrix of $\Gamma$, $\Gamma$ is the weighted adjacency matrix learned from graph G with pair-wise scores as the attention weights, $L^{(i)}$ is the input from previous layer, with $L^{(0)} = \mathbf{U} \in \mathbf{R}^{p \times d}$ as the input matrix corresponding to the embeddings of the source terms, $W_i$ is the learnable weight matrix, $\rho$ is the non-linear activation function.

Note, the end goal of the attentive convolutions is to analyze each word in G in relation to the weighted combination of its semantic neighbors. For this, we surround each word (node) with a group of semantically related words (nodes in the graph) and perform weighted aggregation to recompute the representation of the word. We also allow self-connections for each word, i.e., adding an identity matrix $I$ to $\Gamma$. This will enforce "semantically related words" to get similar representations. We observed, that for our problem settings, this attentive graph convolution framework outperforms the basic settings with equal contribution from all neighbors (Kipf and Welling, 2016).

For GRI, we use a two-layered network to learn

the final embeddings of each word $\mathbf{U}_m \in \mathbf{R}^{p \times d}$ as follows:

$$\mathbf{U}_m = (\tilde{\Gamma})(ReLU((\tilde{\Gamma})\mathbf{U}W_0))W_1 \quad (3)$$

### 4.4 Isomorphism Loss functions

In order to train the GRI, we experiment with multiple different isomorphism loss functions on top of the attentive graph convolution network. Details about each loss function are as follows:

**L2 loss.** We use L2-norm averaged over the number of data samples.

$$\mathcal{L}_2 = \frac{1}{N}||\mathbf{U}_m - \mathbf{V}||_2 \quad (4)$$

**Orthognal Procrustus loss.** Orthogonal Procrustes loss aims to find a linear transformation $W$ to solve:

$$\mathcal{L}_{proc} = \underset{\mathbf{W} \in \mathbf{R}^{d \times d}, \mathbf{W}^T\mathbf{W} = I}{\arg\min} \frac{1}{N}||\mathbf{U}_m\mathbf{W} - \mathbf{V}||_2 \quad (5)$$

Schönemann (1966) proposed a solution $\mathbf{W} = \mathbf{Q}\mathbf{P}^T$, where $\mathbf{P}\Sigma\mathbf{Q}^T$ is the singular value decomposition of $\mathbf{V}^T\mathbf{U}_m$.

**Procrustus loss with initialization.** It follows the same process as that of the Procrustus loss with the exception that we initialize the embeddings for the source words with the embedding vectors of their translation vectors corresponding to the target words. The end goal of this setting is to analyze the ability of the GRI to propagate the knowledge of the initialized embeddings during the model training. We also allow updating the initialized word embeddings during model training. We denote this loss by $\mathcal{L}_{proc_{init}}$. We use the symbol $\mathcal{L}_{ISO}$ to represent the different variations of isomorphism losses, i.e., $\mathcal{L}_2$, $\mathcal{L}_{proc}$ and $\mathcal{L}_{proc_{init}}$.

### 4.5 The Complete Model

Finally, we combine the distributional training objective with the isomorphism loss function to compute complete loss of GRI, as follows:

$$\mathcal{L}_{GRI} = \alpha\mathcal{L}_{SG} + (1-\alpha)\mathcal{L}_{ISO} \quad (6)$$

where $\alpha$ is the parameter used to control the contribution $\mathcal{L}_{SG}$ and $\mathcal{L}_{ISO}$ respectively.

## 5 Experiments and Results

### 5.1 Datasets

In order to set up a unanimous platform for comparative analysis, we use the data settings used by Marchisio et al. (2022). We use the first 1 million lines from newscrawl-2020 data set for English ("en"), Bengali ("bn") and Tamil ("ta") and the entire of newscrawl-2020 data for Ukrainian ("uk") to train word embeddings. We used Moses scripts for data pre-processing[3]. For evaluation, we used publically available train, dev, and test splits provided by MUSE (Conneau et al., 2017). Out of approx 8000-word pairs for each language, we used word pairs 0-5000, 5001-6500, and 6501-8000 as train, test and dev set respectively. The train set is used for model training, dev set is used for parameter tuning, and final results are reported using the test set. All these data splits are non-overlapping.

### 5.2 Baseline Models

For comparative evaluation, we use independently trained distributional embeddings for the source and target languages as a baseline model. Amongst the existing research, we compare GRI against the prevalent state-of-the-art work on BLI, i.e., IsoVec by Marchisio et al. (2022). IsoVec uses the skip-gram training objective along with isomorphism training objectives. Note, Marchisio et al. (2022) used exactly the same data settings as that of our proposed model (i.e., GRI), so for performance comparison, we use the numbers reported in the original paper.

### 5.3 Experimental Settings

For model training, we use Adam optimizer (Kingma and Ba, 2014) with learning rate = 0.001; $\alpha$ in Equation 6 is set to 0.7; the value of $thr$ in algorithm 1 is set to 0.5. For experiments, we use embeddings learnt for English language as the target embeddings, and embeddings for other languages, i.e., "ta", "uk", and "bn", as the source embeddings. For mapping across different spaces, we use Vecmap toolkit with process-flow explained in Section 3.2. We use average P@1 as the evaluation metric. We report mean ($\mu$) and standard deviation ($\sigma$) of the results over 5 runs of the experiment. All experiments are performed using Intel Core-i9-10900X CPU, and Nvidia 3090Ti GPU.

---

[3]Moses script

| Methodology | bn | uk | ta |
|---|---|---|---|
| Baseline | 13.1 ($\pm$ 0.51) | 13.9 ($\pm$ 0.45) | 10.8 ($\pm$ 0.42) |
| IsoVec (L2) | 16.3 ($\pm$ 0.4) | 16.5 ($\pm$ 0.4) | 11.1 ($\pm$ 0.5) |
| IsoVec (Proc-L2) | 16.6 ($\pm$ 0.7) | 16.0 ($\pm$ 0.8) | 10.7 ($\pm$ 0.3) |
| IsoVec (Proc-L2-Init) | 16.9 ($\pm$ 0.2) | 17.1 ($\pm$ 0.6) | 11.8 ($\pm$ 0.3) |
| GRI ($\mathcal{L}_2$) | 17.28 ($\pm$ 0.02) | 18.75 ($\pm$ 0.41) | 13.47 ($\pm$ 0.04) |
| GRI ($\mathcal{L}_{proc_{init}}$) | 19.83 ($\pm$ 0.05) | 21.37 ($\pm$ 0.08) | 15.27 ($\pm$ 0.01) |
| GRI ($\mathcal{L}_{proc}$) | **20.52** ($\pm$ 0.02) | **27.97** ($\pm$ 2.63) | **18.97** ($\pm$ 0.2) |

Table 1: GRI results for the proposed model. We compare results with IsoVec (Marchisio et al., 2022).

## 5.4 Main Results

The results for the proposed model (GRI) compared with the baseline models are shown in Table 1. We boldface the overall best-performing scores with the previous state-of-the-art underlined.

These results show the GRI has a relatively stable performance (with low variance), it consistently outperforms the baseline and previous state-of-the-art scores by a significant margin. For "bn", "uk", and "ta", GRI outperforms the IsoVec (Marchisio et al., 2022) by 21.4%, 63.6% and 60.7% respectively. Especially noteworthy is the performance improvement gained by GRI for the Ukrainian language. We attribute this performance improvement to the fact that the semantic relatedness of the words corresponding to the Ukrainian embedding space is relatively better compared to other languages.

The performance comparison of different isomorphism loss functions shows that $\mathcal{L}_{proc}$ consistently outperforms the $\mathcal{L}_{proc_{init}}$ and $\mathcal{L}_2$ across all data sets. A relatively low performance of $\mathcal{L}_{proc_{init}}$ compared to the $\mathcal{L}_{proc}$ may be attributed to the fact that randomly initialized embeddings are a better choice compared to the initialization from the seed pairs. The initialization from the seed pairs may not be helpful for the model training to improve the performance at later stages.

Overall results show the significance of using attentive graph convolutions in controlling the relative geometry of source language for BLI. Especially, the ability of the attentive convolutions to accumulate the contribution of semantically related terms plays a vital role in controlling the relative geometry of the source embeddings relative to the target embeddings, as is evident from the results in Table 1.

## 6 Discussion

In this sub-section, we perform a detailed analysis of the performance of GRI. Specifically, we analyze: (i) Domain mis-match settings (ii) Impact of attentive convolutions, (iii) Isometric metrics, and

(iv) Error cases.

## 6.1 Domain mismatch

Domain-mismatch has been identified as one of the core limitations of existing BLI methods. These methods fail badly in inferring bilingual information for embeddings trained on data sets from different domains (Søgaard et al., 2018; Marchisio et al., 2020).

We claim that incorporating lexical variations for semantically related tokens makes GRI robust to the domain mismatch settings. In order to validate these claims for GRI, we re-run the experiments using target embeddings trained on 33.8 million lines of web-crawl data from the English Common Crawl data. The embeddings for the source languages ("bn", "uk" and "ta") are trained using the newscrawl-2020 data. The results for the domain-mismatch experiments for different isomorphism loss functions are reported in Table 2.

These results are compared against the baseline distributional embeddings and best-performing scores of the existing work, i.e., IsoVec by Marchisio et al. (2022). Note, for the domain mismatch experiments, we use exactly same data settings as that of Marchisio et al. (2022), so we report exactly the same numbers as reported in original paper. Comparing the results of our model against the IsoVec, the GRI improves the performance by 27.74%, 53.12% and 74.22% for the "bn", "uk" and "ta" languages respectively.

Comparing these results against the main experiments reported in Table 1, we can see the GRI yields a stable performance for both domain-shared as well as domain mismatch settings. These results show that the attentive graph convolutions indeed allow information sharing across semantically related tokens along with their lexical variations that is in turn helpful in controlling the relative isomorphism of the embedding spaces.

Comparing the results for different loss functions, we can see that similar to the main experi-

| Methodology | bn | uk | ta |
|---|---|---|---|
| Baseline | 9.7 ($\pm$ 0.72) | 10.2 ($\pm$ 0.43) | 7.5 ($\pm$ 0.39) |
| IsoVec (Proc-L2-Init) | 15.5 ($\pm$ 0.7) | 17.3 ($\pm$ 0.4) | 10.9 ($\pm$ 0.5) |
| GRI ($\mathcal{L}_2$) | 13.97 ($\pm$ 0.02) | 17.32 ($\pm$ 0.32) | 11.93 ($\pm$ 0.01) |
| GRI ($\mathcal{L}_{proc_{init}}$) | 19.75 ($\pm$ 0.01) | 21.32 ($\pm$ 0.10) | 17.12 ($\pm$ 0.59) |
| GRI ($\mathcal{L}_{proc}$) | **19.80** ($\pm$ 0.50) | **26.49** ($\pm$ 0.50) | **18.99** ($\pm$ 0.20) |

Table 2: GRI results for domain mis-match experiments compared with the baseline models, IsoVec (Marchisio et al., 2022).

| Methodology | bn | uk | ta |
|---|---|---|---|
| GRI w/o G-Conv ($\mathcal{L}_2$) | 16.10 ($\pm$ 0.35) | 16.35 ($\pm$ 0.30) | 11.25 ($\pm$ 0.45) |
| GRI w/o G-Conv ($\mathcal{L}_{proc_{init}}$) | 16.75 ($\pm$ 0.20) | 16.98 ($\pm$ 0.30) | 11.70 ($\pm$ 0.25) |
| GRI w/o G-Conv ($\mathcal{L}_{proc}$) | 16.50 ($\pm$ 0.5) | 16.10 ($\pm$ 0.70) | 10.65 ($\pm$ 0.20) |
| GRI ($\mathcal{L}_2$) | 17.28 ($\pm$ 0.02) | 18.75 ($\pm$ 0.41) | 13.47 ($\pm$ 0.04) |
| GRI ($\mathcal{L}_{proc_{init}}$) | 19.83 ($\pm$ 0.05) | 21.37 ($\pm$ 0.08) | 15.27 ($\pm$ 0.01) |
| GRI ($\mathcal{L}_{proc}$) | **20.52** ($\pm$ 0.02) | **27.97** ($\pm$ 2.63) | **18.97** ($\pm$ 0.20) |

Table 3: Analyzing the impact of attentive graph convolutions for GRI.

ments the performance of the model for the Procrustes loss ($\mathcal{L}_{proc}$) is relatively higher than the $\mathcal{L}_2$ and $\mathcal{L}_{proc_{init}}$.

## 6.2 Impact of attentive convolutions

In this sub-section, we analyze in detail the performance improvement of GRI attributable to the attentive graph convolutions. For this, we primarily analyze the performance improvement of GRI with and without attentive graph convolutions. The results of these experiments are reported in Table 3. These results show the significance of attentive graph convolutions that help in improving the performance across all three languages. The improvement in performance for the "bn", "uk" and "ta" languages is 24.36%, 64.72% and 62.13% respectively.

To gain further insight, we also analyzed the output of the model with and without graph convolutions in order to dig out which class of translation instances were correctly translated especially due to the attentive convolutions part of GRI. We run this analysis only for the Ukrainian language because: GRI yields a higher score for the Ukrainian language compared to other languages. All the analyses were performed under the direct supervision of a linguistic expert.

Detailed analyses show that a major portion (approx 51%) of the pairs corrected especially by the graph convolutions belong to the nouns, with 21% verbs and 20% adjectives. The rest 7% are assigned to other classes. This analysis shows that the phenomenon of lexical variation is dominant among nouns that results in better performance of GRI compared to the baseline models.

## 6.3 Isometric metrics

We also correlate the results of GRI with different widely used isomorphism metrics. Specifically, we use two metrics, namely: (a) Pearson's correlation, and (b) Eigenvector similarity. Details about these metrics and the corresponding experimental setting are as follows:

**Pearson's Correlation.** We compute Pearson's correlation between the cosine similarities of the

| | Pearson Correlation ($\uparrow$) | | | Eigenvector Similarity($\downarrow$) | | |
|---|---|---|---|---|---|---|
| Methodology | bn | uk | ta | bn | uk | ta |
| GRI ($\mathcal{L}_2$) | 0.47 | 0.36 | 0.42 | 35.55 | 30.64 | 69.72 |
| GRI ($\mathcal{L}_{proc_{init}}$) | 0.47 | 0.36 | 0.43 | **31.23** | **10.92** | **45.56** |
| GRI ($\mathcal{L}_{proc}$) | **0.49** | **0.37** | **0.44** | 32.16 | 29.53 | 62.81 |

Table 4: Analysis of different isometry metrics for GRI.

seed translation pairs as an indicator of the relative isomorphism of corresponding spaces. We expect our P@1 results to correlate positively ($\uparrow$) with Pearson's correlation.

We compute Pearson's correlation over first 1000 translation seed pairs. Corresponding results are shown in the first half of Table 4. We boldface the best scores. These results show that for all languages, Pearson's correlation for the model GRI ($\mathcal{L}_{proc}$) is slightly higher compared to other models. Although these results are aligned with our findings in Table 4, however, one noteworthy observation is that Pearson's correlation is not a true indicator of the relative performance improvement across different isomorphism losses.

**Eigenvector Similarity.** In order to compute the eigenvector similarity of two spaces, we compute the Laplacian spectra of the corresponding k-nearest neighbor graphs. This setting is similar to Søgaard et al. (2018), and is summarized as follows. For seed pairs construct unweighted graphs followed by computing the graph Laplacians. Later, compute the eigenvalues of the graph Laplacians and retain the first $k$ eigenvalues summing to less than 90% of the total sum of eigenvalues. Finally, we compute the eigenvector similarity as the sum of squared differences between partial spectra. The graphs with similar eigenvalue spectra are supposed to have similar structures (a measure of relative isomorphism).

We expect our eigenvector similarity results to correlate negatively ($\downarrow$) with P@1. The experimental results are shown in the right half of Table 4, with the best scores boldfaced. These results show that the eigenvector similarity scores for the model GRI ($\mathcal{L}_{proc_{init}}$) are better than the other two models. This is in contrast to our findings in Table 1, where GRI ($\mathcal{L}_{proc}$) shows relatively better performance.

Generally speaking, the results of the isometric metrics do not truly correlate with the P@1. These findings are aligned with earlier studies by Marchisio et al. (2022) that also emphasized the need for better metrics to compute the relative isomorphism of the embedding spaces.

## 6.4 Error Analyses

We also analyze the error cases of GRI in order to know the limitations of the model and room for future improvement. Note, similar to section 6.2, we only perform the error analyses for the Ukrainian language and Procrustes loss ($\mathcal{L}_{proc}$). All experiments were performed with the help of linguistic experts. We separately analyze the errors for the variants of GRI with and without attentive graph convolutions (i.e., GRI; GRI w/o G-Conv) in order to quantify the reduction in error attributable to the attentive graph convolutions.

In order to better understand the errors from semantic perspective, we categorize the errors into the following types:

**Type-a:** The predicted target word for P@1 is semantically close to the true target word.

**Type-b:** The predicted target word is a k-nearest neighbor of the true word for k=5.

We limit the error cases to only the above-mentioned simple types because these types give a rough picture of the relative isomorphism of the different spaces from the semantic perspective. The percentage error counts for both models are shown in Table 5. For the model GRI w/o G-Conv($\mathcal{L}_{proc}$), 21.3% errors fall in error Type-a, and 6.5% errors belong to error Type-b. For the model GRI($\mathcal{L}_{proc}$), 50.2% errors fall in Type-a, and 16.6% errors belong to Type-b. As expected the variant of GRI with graph convolutions shows a higher percentage for both categories, i.e., Type-a and Type-b. These numbers clearly indicate that the attentive graph convolutions were not only able to correct a major portion of errors made by (GRI w/o G-Conv), but also the errors made by the model GRI are either highly semantically related to the target words or a nearest neighbor of the target word.

In order to gain further insight, we manually look at the error cases. For both models, a few examples are shown in Table 6. The majority of the predictions made by GRI are indeed correct and closely related to the true target words. For example, it predicts {"mailing", "sharing", "windows"} in place of {"mail", "shared", "window"}

| | Type-a | Type-b |
|---|---|---|
| GRI w/o G-Conv($\mathcal{L}_{proc}$) | 21.3% | 6.5% |
| GRI ($\mathcal{L}_{proc}$) | 50.2% | 16.6% |

Table 5: Classification of Error Types

| GRI ($\mathcal{L}_{proc}$) | | | GRI w/o G-Conv($\mathcal{L}_{proc}$) | | |
|---|---|---|---|---|---|
| source | target | target$'$ | source | target | target$'$ |
| пошта | mail | mailing | зникли | gone | shattered |
| спільний | shared | sharing | олія | oil | 60g |
| вікно | window | windows | банки | cans | merchants |
| внз | college | teaching | ніс | nose | rubbing |
| йшов | walked | went | заміна | replacing | overpriced |
| підручників | manuals | templates | вулкан | volcano | 100mph |
| реформа | reform | reforms | ріст | growth | decline |

Table 6: Example error cases for the Ukrainian vs English language for the models: GRI ($\mathcal{L}_{proc}$); GRI w/o G-Conv($\mathcal{L}_{proc}$). For each model, the first column (source) corresponds to the Ukrainian words, the second column (target) represents the true target word, third column (target$'$) represents the model predictions for target words.

respectively. These results clearly indicate that the current performance of GRI is under-reported and there is a need for better quantification measures (other than P@1) in order to compute and/or report the true potential of the model.

Overall error analyses show the significance of the using attentive graph convolutions to incorporate the lexical variations of semantically related tokens in order to control the relative isomorphism and perform BLI in performance-enhanced way.

## 7 Conclusion and Future Directions

In this paper, we propose Graph-based Relative Isomorphism (GRI) to incorporate the impact of lexical variations of semantically related tokens in order to control the relative isomorphism of cross-lingual embeddings. GRI uses multiple different isomorphism losses (guided by the attentive graph convolutions) along with the distributional loss to perform BLI in a performance-enhanced fashion. Experimental evaluation shows that GRI outperforms the existing research on BLI by a significant margin. Some probable future directions include: (i) extending the concepts learned in this research to contextualized embeddings, and (ii) augmenting the GRI to focus more on preserving lexico-semantic relations.

## Limitations

Some of the core limitations of the proposed approach are as follows: (i) current formulation of GRI is not defined and/or implemented for deep contextualized embeddings which are more prevalent and a better alternate to the distributional embeddings, (ii) existing limitations of the distributional embeddings are inherited in the model, which limits the end-performance of GRI. For example, as pointed out by Ali et al. (2019) the distributional embedding space tends to inter-mix dif-

ferent lexico-semantic relations, and yield a poor performance on a specific task. This phenomenon has a direct impact on GRI especially on controlling the relative isomorphism of highly interlinked relation pairs, e.g., Antonyms vs Synonyms.

**Acknowledgements.** Di Wang, Yan Hu and Muhammad Asif Ali are supported in part by the baseline funding BAS/1/1689-01-01, funding from the CRG grand URF/1/4663-01-01, FCC/1/1976-49-01 from CBRC and funding from the AI Initiative REI/1/4811-10-01 of King Abdullah University of Science and Technology (KAUST). Di Wang is also supported by the funding of the SDAIA-KAUST Center of Excellence in Data Science and Artificial Intelligence (SDAIA-KAUST AI).

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
