# OpenReview forum: "GRI: Graph-based Relative Isomorphism of Word Embedding Spaces"
_EMNLP/2023/Conference — EMNLP 2023 Findings_

### Official Review · Reviewer_spUb · 2023-08-01

**Soundness:** 4

**Excitement:**

3: Ambivalent: It has merits (e.g., it reports state-of-the-art results, the idea is nice), but there are key weaknesses (e.g., it describes incremental work), and it can significantly benefit from another round of revision. However, I won't object to accepting it if my co-reviewers champion it.

**Paper Topic And Main Contributions:**

This work addresses the challenge of building bilingual dictionaries for machine translation, which is  affected by the alignment of semantically related terms in each language. Prior methods have effectively combined the geometric spaces across languages but have not considered the impact of semantically related terms. This work thus proposes a method called GRI (Graph-based Relative Isomorphism), which is supposed to account for the impact of semantically similar terms using attentive graph convolutions. This is enables effective computation of the relative isomorphism of multiple geometric spaces. The results indicate that the GRI performs existing methods by using the average precision at rank 1 measure, demonstrating that GRI leads to more accurate alignment between terms of different languages.

**Reasons To Accept:**

The contribution of this paper seems to be significant in that the GRI method is proposed, leading to improvements in bilingual dictionary creation and thus in machine translation. The paper address an important problem in machine translation and demonstrates a performance improvement via P@1 of 63.6%. The authors also demonstrate that these results hold in domain mismatch settings, which is a topical concern.

**Reasons To Reject:**

The methodology and contributions were not very clear to me, especially when comparing this method against straight BLI, but that is in part because this is not my area. It is also not clear whether these results are very reproducible and how many runs were performed for each set of results.

**Reproducibility:**

3: Could reproduce the results with some difficulty. The settings of parameters are underspecified or subjectively determined; the training/evaluation data are not widely available.

**Reviewer Confidence:**

1: Not my area, or paper was hard for me to understand. My evaluation is just an educated guess.

**Typos Grammar Style And Presentation Improvements:**

Incorrect citation for Søgaard 2018-- the same paper appears twice in the Bibliography (once as 2018a and once as 2018b)

Incorrect citation for Conneau 2017-- the same paper appears twice in the Bibliography (once as 2017a and once as 2017b)

Many papers which were cited as arxiv papers have since been peer-reviewed and published; cite the peer-reviewed versions

---

> ### Author Rebuttal · Authors · 2023-08-28
>
> Thanks for your valuable feedback. We will consider these points while preparing the final draft of the paper. The response for each individual point raised in the paper is as follows:
>
> ### Contributions of GRI:
>
> 1. The main contribution of GRI is to propose Attentive Graph Convolutions to incorporate the impact of "lexically varying but semantically similar words" on Bilingual Lexical Induction (BLI).
>
> 2. In order to better highlight the main contribution of the paper, we will additional intuitive explanations throughout the paper. We will also add concrete examples in the introduction (in addition to the current example in Section 4.2) to emphasize the impact of Attentive Graph Convolutions compared to existing BLI.
>
> ### Reproducibility:
>
> 1. All the methodologies developed in the paper and corresponding results have been computed using freely available public data resources, as stated in Section 5.1 of the paper. These results are easily reproducible. Moreover, upon acceptance, we will make all the codes and data sets publicly available to ensure easy reproducibility of the results.
>
> 2. All results reported in the paper are averaged over 5 runs of the experiments as stated in lines 403 to 406 of the paper.
>
> ### Typos:
>
> 1. All the bibliographic references including (Søgaard et al., (2018), Conneau et al., (2017)) have been corrected to remove duplicate entries.
>
> 2. All the arxiv references have been replaced by corresponding peer-reviewed versions.

---

### Official Review · Reviewer_AjeL · 2023-08-03

**Soundness:** 3

**Excitement:**

3: Ambivalent: It has merits (e.g., it reports state-of-the-art results, the idea is nice), but there are key weaknesses (e.g., it describes incremental work), and it can significantly benefit from another round of revision. However, I won't object to accepting it if my co-reviewers champion it.

**Paper Topic And Main Contributions:**

The authors propose a new graph isomorphism loss function that is used in combination with the skip gram loss function to improve the relative isomorphism of static word embedding spaces, therefore improving their ability to be mapped to a shared cross-lingual space as measured by P@1.  This work extends Marchisio et al., (2022) (IsoVec) to a new loss function.

**Questions For The Authors:**

Lines ~240-244 - why does semantic relatedness imply that graphs are a natural choice?
4.3.1 - did you implement SkipGram yourself?
Which retrained word2vec embedding did you use?  Note that MUSE is fasttext.
“We aggregate the information contained in the neighbors to come up with a new representation for the word that accommodates information from semantically related neighbors” - how?
Section 4.3.2 - the math notation is confusing. What is the encircled dot?  What is n_ij?  I recommend explaining only what is necessary, with no jargon or unnecessary equations, as these obscure what you are doing.
(6) - I do not see that L_iso is defined.
Line 402 - “process flow”?

**Reasons To Accept:**

Marchisio et al. (2022) called for the community to create additional loss functions to improve the relative isomorphism of word embedding spaces, and this is one such metric.  The performance vs. the aforementioned work are strong.

**Reasons To Reject:**

My biggest criticism is that I do not feel that the method is explained clearly and concisely.  The explanation in 4.3 seems obscured by unnecessary and unclear math with non-standard notation.  I would be more enthusiastic if the method were explained clearly and concisely, as it is difficult to tell what exactly is happening.

**Reproducibility:**

3: Could reproduce the results with some difficulty. The settings of parameters are underspecified or subjectively determined; the training/evaluation data are not widely available.

**Reviewer Confidence:**

4: Quite sure. I tried to check the important points carefully. It's unlikely, though conceivable, that I missed something that should affect my ratings.

**Typos Grammar Style And Presentation Improvements:**

Bi-lingual is typically written as “bilingual”
IsoVec is the correct captialization
A concrete example in the intro of what is meant by “lexically different but semantically similar” would be helpful
Line 152 is missing a year for the Marchisio citation
Figure 1 could be better explained
line 181 - I actually do not think that dimensionality reduction happens happens.  I would check the implementation you can.
Line 196: extend->extent

---

> ### Author Rebuttal · Authors · 2023-08-28
>
> Firstly, thanks for a detailed read of the paper. It will significantly help in preparing the final draft of the paper if accepted. The response for each individual point is as follows:
>
> ### Contributions of GRI:
>
> 1. The main contribution of GRI is the use of Attentive Graph Convolutions to incorporate the impact of "lexically varying but semantically related words", {e.g., terrible, horrible, awful, etc.,} for Bilingual Lexical Induction (BLI). The key idea is to surround each word (node in the graph) with a group of semantically related words (though lexically different) and perform weighted aggregation to re-compute the representation of the word. This will enforce "lexically varying but semantically related words" to be embedded close to each other, which in turn boosts the end performance of the BLI systems. We will add more details and concise explanations throughout the paper (especially Section 4.3) to make it easy to understand.
>
> ### Answers to Questions:
>
> 1. **Line 240-244:** It is not the semantic relatedness between words, instead, it is the computations over an arbitrary number of lexical variations for words that are semantically related to a given word that make graphs a natural choice. Details are as follows:
>
>     * In order to capture multiple lexical variations of a given word, GRI first constructs a graph such that each word (node in the graph) is surrounded by an arbitrary number set of semantically related words (i.e., neighbors in the graph).
>
>     * Later, re-compute the representation of each word by aggregating the information from neighbors over multiple hops. Note it also provides the provision to aggregate the information from immediate and higher-order neighborhoods by stacking multiple layers.
>
>
> 2. **Skip-Gram:** Yes, we implemented the skip-gram model using Pytorch. Our implementation of skip-gram is primarily inspired by existing works, i.e., online tutorials, skip-gram with negative sampling [1], and IsoVec [6].
>
> 3. **Word2Vec Embeddings:** For Graph construction, we use existing Word2Vec Embeddings pre-trained using Google-News Corpus of 100 billion words.
>
> 4. **MUSE:** Yes, MUSE (Multi-lingual Unsupervised and Supervised Embeddings) provides the provision for multilingual embeddings (i.e., fastText embeddings aligned in a common space). However, MUSE learns mapping functions to map independently trained embeddings to a common space, which has severe limitations, e.g.,
>
>     * It relies on the assumption of geometric similarity of underlying mono-lingual embeddings.
>
>     * It performs poorly in domain-mismatch settings as shown by existing works [3, 5]. This is also stated in lines (32 to 51) of the paper.
>
> 5. **New Representation:** The key idea is to surround each word (node) with a group of semantically related words (nodes in the graph) and perform weighted aggregation to re-compute the representation of the word. This will enforce "lexically varying but semantically related words" to get similar representations.
>
> 6. **Maths Notation:** The mathematical notation used in section 4.3.2 is inspired by earlier work on Graph Convolution by Kipf and Welling [2] with very few additions. We will work out the ambiguous parts to make it more clear.
>
> 7. **Encircled dot:** The encircled dotted is used to represent the Hadamard product, i.e., the element-wise product of the attention matrix $\zeta$ and the adjacency matrix A.
>
> 8. **$\eta_{ij}$:** $\eta_{ij}$ is a typo. It has been corrected and replaced with $\zeta$ to represent the attention matrix, and $\zeta_{ij}$ to represent the attention weight between the words $w_i$ and $w_j$.
>
>     * To re-emphasize, we use a multi-layer network with the following layer-wise propagation mechanism to compute the representation for layer $L^{(i+1)}$ given $L^{(i)}$: $L^{(i+1)} = \rho(\widetilde{(\zeta \odot A)} L^{(i)} W_{i})$,
>
>     * Here $\rho$ and $W_{i}$ are the non-linearity and weight matrix respectively. $L^{(i)}$ is the representation for the i-th layer.
>
>     * $\widetilde{\zeta_{} \odot A}= \bar{D}^{-1/2} ((\zeta_{} \odot A) + I) \bar{D}^{-1/2}$ is the normalized symmetric matrix with $\bar{D}$ as the degree matrix of $(\zeta_{} \odot A)$. Details about the normalization trick are available in [2]. We will make it more clear in the paper.
>
> 9. **$L_{ISO}$:** We use three different variations for $L_{ISO}$ (explained in line 331 of the paper), i.e., $L_2$, $L_{porc}$, and $L_{{porc}_{init}}$. These are explained in detail in the Section 4.4 of the draft.
>
> 10. **Line 402:** It states that the process flow of the Vecmap toolkit [4] for mapping across different spaces is explained in Section 3.2.
>
> ### Improvements + Typos:
>
> 1. We have updated the word bi-lingual to "bilingual" throughout the document.
>
> 2. We have replaced all instances of  "Isovec" with its correct capitalization: "IsoVec".
>
> 3. Currently, we discuss examples of lexically different but semantically similar words, {i.e., "terrible", "horrible", "awful" etc.,} in Section 4.3. We will also add a solid and concrete example in the introduction along with the essence of modeling them using a graph. Thanks for mentioning it.
>
> 4. Line 152 -- Marchisio's citation has been updated to Marchisio et al. (2022).
>
> 5. For the final draft, we will add more details for a better explanation of Fig 1, i.e., skip-gram followed by Attentive Graph Convolutions and finally combining skip-gram and isomorphism loss. We will make the overall workflow of GRI more explicit and self-explainable.
>
> 6. Line 181 explains the process flow of the Vecmap toolkit [4]. Dimensionality reduction is an optional step in Vecmap. In our case, it does not significantly impact the final results. We will omit this step from the final draft of the paper.
>
> 7. Line 196: the word "extend" has been updated to "extent".
>
>
>
> ### References:
>
> 1. Tomas Mikolov, Ilya Sutskever, Kai Chen, Greg Corrado, and Jeffrey Dean. "Distributed Representations of Words and Phrases and their Compositionality." NIPS (2013).
>
> 2. Kipf, Thomas N., and Max Welling. "Semi-supervised classification with graph convolutional networks." ICLR (2016).
>
> 3. Anders Søgaard, Sebastian Ruder, and Ivan Vulic. "On the limitations of unsupervised bilingual dictionary induction." ACL (2018).
>
> 4. Artetxe, M., Labaka, G., and Agirre, E. "Generalizing and Improving Bilingual Word Embedding Mappings with a Multi-Step Framework of Linear Transformations." AAAI (2018).
>
> 5. Goran Glavas, Robert Litschko, Sebastian Ruder, and Ivan Vulic. "How to (properly) evaluate cross-lingual word embeddings: On strong baselines, comparative analyses, and some misconceptions." ACL (2019).
>
> 6. Kelly Marchisio, Neha Verma, Kevin Duh, and Philipp Koehn. "IsoVec: Controlling the Relative Isomorphism of Word Embedding Spaces." EMNLP (2022).

---

### Official Review · Reviewer_oHhd · 2023-08-04

**Soundness:** 4

**Excitement:**

4: Strong: This paper deepens the understanding of some phenomenon or lowers the barriers to an existing research direction.

**Paper Topic And Main Contributions:**

This paper proposes a novel graph convolution approach for Bilingual Lexicon Induction (BLI).
The graphs used in the work captures relations between semantically related words (e.g. "good", "awesome") and the approach tries to minimize graph isomorphism in bilingual spaces.

The experiments are very comprehensive, in particular in (1) reporting 5 random trials for each of the three language pairs, (2) conducting different data setups, such as the more challenging but realistic domain mismatch setup in Sec 6.1, and (3) ablation experiments and error analyses in Sec 6.2-6.4. In general, I think this is a nice paper that will help push the frontier of BLI.

**Questions For The Authors:**

- Sec 4.3 Do you have multiple cliques in the graph? Do you get a fully-connected graph across all words? I understand it's a fully-connected graph between each group of semantically similar words but I am wondering if there are connections between those that aren't semantically similar according to your threshold.



**Reasons To Accept:**

- Novel Technique: the emphasis on semantically similar words is new and the use of graph convolutions is quite interesting.
- Comprehensive experiments
- Strong results

**Reasons To Reject:**

none in particular

**Reproducibility:**

4: Could mostly reproduce the results, but there may be some variation because of sample variance or minor variations in their interpretation of the protocol or method.

**Reviewer Confidence:**

4: Quite sure. I tried to check the important points carefully. It's unlikely, though conceivable, that I missed something that should affect my ratings.

**Typos Grammar Style And Presentation Improvements:**

- Fig 1 and Attentive Graph Convolutions subsection in Sec 4.3 - I think it might be helpful to also  try to explain intuitive what does your graph method do. It may not be very clear to some how the graph attention interacts with your graph construction to create isomorphic results.

Typos:
- Sec 4.2 "best possible extend" -> extent
- Sec 4.3 english and ukrainian languages. -> uppercase language name

---

> ### Author Rebuttal · Authors · 2023-08-28
>
> Thanks for your valuable feedback. It will help a lot in improving the overall content of the paper. Our response for each individual point is as follows:
>
> ### Answer to Question:
>
> 1. Yes, depending on the value of the threshold we end up with an adjacency graph with multiple cliques. There are no connections between the node pairs with similarity lower than the threshold values, as it will incur noise in the model training.
>
>
> ### Presentation improvement:
>
> 1. For Figure 1, we will add more details to the figure's content and its explanation in order to make the overall workflow of GRI more explicit and self-explainable.
>
> 2. For Section 4.3, the simplest intuitive explanation of the Attentive Graph Convolutions is to perform a weighted aggregation of lexically varying but semantically related words. This in turn enforces the representation of these words to be close to each other for effective BLI.
>
> 3. The simplest intuitive explanation for the interaction of the graph construction and attention mechanism is as follows:
>
>     * Graph construction aims at formulating the graph in a way that: "lexically different but semantically similar words end up being neighbors in the graph".
>
>     * Later, Attentive Graph Convolutions aggregate the information in the graph to re-compute the representation of each word in the graph. We will these details in the final draft to make it more clear.
>
> ### Typos:
>
> 1. In Sec 4.2 "best possible extend" has been updated to: "best possible extent".
>
> 2. All language names (Ukrainian, Tamil, Bengali, and English) have been upper-cased throughout the draft.

---

### Meta-Review · Area_Chair_Su2P · 2023-09-19

**Recommendation:** 3

**Metareview:**

This paper proposes a graph convolution approach for the Bilingual Lexicon Induction task. Overall, the reviewers found that the paper presents interesting observations and promising experimental results. Strong experimental results over comparable and the latest method in the field of bilingual lexical induction with the graph isomorphism-inspired method such as IsoVec (Marchiso et al. 2022). However, multiple reviewers pointed out concerns about the clarity of the submission (e.g., math notation [1, 2], lack of definitions of some phrases). The AC would also strongly encourage adding technical details and motivation for selecting components (e.g., why using Graph Convolutional Network but not other graph NN models) but make the paper clear and concise. The AC strongly recommends addressing detailed comments made by Reviewer AjeL to improve the next version of the paper for future readers.


[1] In addition to what reviewers recommended for increasing the clarity on math notations, the AC would also suggest using different styles for loss $L$ vs. layer $L$ e.g., by using $\mathcal{L}$. These small changes pile up and save a lot of mental effort for future readers.

[2] E.g., the definition of $L_{ISO}$ is not clear from the main text, and adding the explanations mentioned in the rebuttal https://openreview.net/forum?id=IsDxBXUEd8&noteId=FV6vFXbdKa would definitely benefit future readers to easily understand the paper.

---

### Decision · Program_Chairs · 2023-10-07

**Decision:**

Accept-Findings

**Comment:**

This paper proposes a graph convolution approach for the Bilingual Lexicon Induction task. Overall, the reviewers found that the paper presents interesting observations and promising experimental results. Strong experimental results over comparable and the latest method in the field of bilingual lexical induction with the graph isomorphism-inspired method such as IsoVec (Marchiso et al. 2022). However, multiple reviewers pointed out concerns about the clarity of the submission (e.g., math notation [1, 2], lack of definitions of some phrases). The AC would also strongly encourage adding technical details and motivation for selecting components (e.g., why using Graph Convolutional Network but not other graph NN models) but make the paper clear and concise. The AC strongly recommends addressing detailed comments made by Reviewer AjeL to improve the next version of the paper for future readers.


[1] In addition to what reviewers recommended for increasing the clarity on math notations, the AC would also suggest using different styles for loss $L$ vs. layer $L$ e.g., by using $\mathcal{L}$. These small changes pile up and save a lot of mental effort for future readers.

[2] E.g., the definition of $L_{ISO}$ is not clear from the main text, and adding the explanations mentioned in the rebuttal https://openreview.net/forum?id=IsDxBXUEd8&noteId=FV6vFXbdKa would definitely benefit future readers to easily understand the paper.